# Comparison of the Characteristics of Recycled Carbon Fibers/Polymer Composites by Different Recycling Techniques

**DOI:** 10.3390/molecules27175663

**Published:** 2022-09-02

**Authors:** Kwan-Woo Kim, Dong-Kyu Kim, Woong Han, Byung-Joo Kim

**Affiliations:** 1R&D Office 1st, Korea Carbon Industry Promotion Agency, Jeonju 54852, Korea; 2Department of Carbon-Nanomaterials Engineering, Jeonju University, Jeonju 55069, Korea

**Keywords:** carbon fibers, recycling, upcycling, recovery, carbon fibers reinforced plastics

## Abstract

In this study, three recycling methods, namely, mechanical grinding, steam pyrolysis, and the supercritical solvent process, which are used to acquire recycled carbon fibers (RCFs), were compared for their application in synthesizing polymer-matrix composites. RCF-reinforced polyethylene (PE) composites were prepared to compare the mechanical properties of the composites generated using the three recycling methods. The PE/RCF composites exhibited 1.5 times higher mechanical strength than the RCF-reinforced PE composites, probably because of the surface oxidation effects during the recycling processes that consequently enhanced interfacial forces between the RCF and the matrix. Further, the steam pyrolysis process showed the highest energy efficiency and can thus be applied on a large production scale in domestic recycled CF markets.

## 1. Introduction

Carbon-fiber-reinforced plastic (CFRP) is an advanced composite material having high strength, high elasticity, and extremely low weight. These properties have promoted its widespread application in various fields, such as automobiles, sports, aerospace, and the military [1,2,3,4,5,6]. However, the large-scale use of CFRPs generates vast quantities of waste. Generally, thermoplastic and thermosetting resins are the most commonly used matrixes in CFRPs. CFRPs with thermoplastic resins can be easily recycled because they are reversible. However, the separation of carbon fibers (CFs) from the matrix of thermosetting-based CFRPs is difficult, resulting in their disposal in landfills after incineration. This disposal type makes it difficult to recycle CFs; additionally, this process is costly and causes environmental pollution. Thus, the demand for waste recycling has been increasing [7,8,9].

Various recycling methods for waste CFRPs have been studied to reuse expensive CFs [10]. The main recycling methods are mechanical recycling [11,12], chemical processes [13,14], and thermal processing [15,16,17]. Among these, mechanical recycling is the simplest and the most cost-effective process; moreover, although CFs can be recovered from waste CFRP through grinding, crushing, and cutting, only pure CFs cannot be obtained [18]. The chemical process recycles CFs through decomposition using a solvent [19,20], a supercritical method [21,22], etc.; moreover, recycling through this process can be conducted while retaining the mechanical properties of CFs as much as possible. However, the continuous application of the chemical process for recycling CFs releases extensive amounts of harmful gases and generates toxic reagents, which is a major disadvantage [23]. Lastly, thermal processing, which is the most widely used recycling process for CFRPs, can recover relatively clean CFs by thermally decomposing organic molecules into smaller molecules. In addition, the mechanical properties of the CFs can be retained through this method (a tensile strength of 70–80% compared to virgin CF), thereby making it a suitable alternative for large-scale commercialization [23,24,25].

Recently, in European Union countries, upcycling, which is one step beyond the simple recycling process, has been used extensively in industrial applications. It is also attracting attention in several countries, such as the United States and Canada, as an eco-friendly production and ethical consumption method [26,27]. Because of its environmental and economic advantages, studies on the upcycling and recycling of CFRPs are increasing.

This study aimed to investigate mechanical recycling, supercritical recycling, and superheated steam recycling processes to compare the application characteristics of CF composites recycled by these methods. CFs recycled through each process were mixed with low-density polyethylene (LDPE) to form a composite to assess the mechanical, electrical, and thermal properties of the composites. Further, the application of surface treatment to CFs can result in the recovery of clean CFs and maintain their physical properties during recycling; additionally, the interfacial properties with the matrix can be improved. Therefore, upcycling waste CFRPs can further increase the resource value of the generated CFRP wastes.

## 2. Results and Discussion

### 2.1. Characteristics of the CFRPs Acquired through the Studied Recycling Techniques

Figure 1 shows the SEM images of CFRP scrap waste, virgin CFs, supercritical recycled CFs, and superheated steam recycled CFs.

In the CFRP scrap waste, a large amount of resin was wrapped around the CF, and epoxy resin fragments broken during the cutting process were observed. Although some resin decomposition products remained on the recycled CF surface during the supercritical and superheated steam recycling methods, virgin CFs were recovered relatively effectively, as shown in the SEM image (Figure 1). Further, recycled CFs recovered from superheated steam showed relatively cleaner surfaces than those recovered by mechanical and supercritical recycling methods.

### 2.2. Physical Characteristics of the Various RCF-Reinforced Composites

The results of the Charpy pendulum impact test for the LDPE matrix CFRPs with different mixed fillers are presented in Figure 2.

The specific Charpy energy of the PE/SHS-RCF CFRPs was evidently higher than that of the PE/CF CFRPs. This could be attributed to the increase in the interfacial shear strength (IFSS), which improved the high-impact resistivity between the LDPE and CFs. Table 1 shows the results of previous studies [23] conducted on recycled CFs and untreated CFs under the same conditions as used in this study.

The average IFSS value for the CFs was 39.19 MPa. Further, the IFSS values of the RCFs were enhanced (47.06 MPa) after the surface pyrolysis of the CFRPs using SHS. Pyrolysis (SHS) oxidation introduces –OH, –CH–, H-bond, –C=O, and C–O functional groups successfully onto the fiber surface. The –OH or –COOH groups can form covalent interfacial bonds in the cross-linked polymer adhesive which couple to the fiber, effectively transferring stresses between the matrix and the fiber and improving interfacial adhesion. Furthermore, fracture toughness is improved by hydrogen and covalent bonding interactions. Surface functionalization increases polar components and the total surface free energy of the fibers. Moreover, the RCF surface treated with SHS contains hydrophilic functional groups, such as oxygen carboxyl, carbonyl, and hydroxyl. The high polar term in the total surface free energy can be expected to contribute to the good wettability and adhesion between the fiber and matrix. In the case of SC-RCF, reliable surface free energy values could not be obtained due to many impurities on the surface.

In this study, the specific Charpy energy reached 36.64 kJ/m^2^ for the PE/SHS-RCF, which meets the standard demands of engineered materials. This improved impact strength is believed to be due to the increased oxygen content after the surface pyrolysis of CFRPs with SHS.

As shown in Figure 3, the morphologies of the fractured surfaces of the four samples differ. Closer examination reveals that the fractured surface morphologies of (a) to (d) differ in roughness as well. The fractured surface of (c) differs from that of (a) and (b) because of the stronger adhesion between the CFs and the matrix in (d). As shown in (a), a smooth surface with marginally adhered resin can be seen on the CF surface in the debonded area; additionally, the matrix can be observed to have been detached from the fiber surface. This indicates cohesive failure at the interface between the matrix and CFs. Further, the absence of residual LDPE on the CF surface is direct evidence of a weak interface [28,29]. Figure 3d shows that the interface is closely connected to the resin due to having a higher bonding force than the samples (a) and (b). However, in the vicinity of the fiber, the resin shape was stretched by the impact. Consequently, sample (c), with no shape change at the interface between the fiber and the resin, showed the highest interfacial bonding force.

### 2.3. Thermal and Electrical Properties of RCF-Reinforced Composites

Table 2 enlists the horizontal thermal and electrical conductivity values of the CFRPs, which showed high impact strength.

The low thermal conductivity of LDPE (approximately 0.1 to 0.3 W/mK) can be enhanced by the filler effect (1 to 2 W/mK in the horizontal direction). The thermal conductivity of the PE/SHS-RCF sample in the horizontal direction was observed to be slightly higher than that of PE/CF, but it was judged to be an approximate value. This is because the CFs are well-dispersed during the molding and production of the composite material; additionally, the plurality of the fillers is oriented in the horizontal direction by the flow of the molten resin during hot compression. Because the effect of CFs on long fibers with high thermal conductivity is dominant, the degree of dispersion in CFs due to interfacial bonding force may result in the loss of internal heat transfer path, thereby subsequently affecting thermal conductivity (Figure 4).

Further, the results indicate that the high resistivity of LDPE (approximately 10^15^–10^18^ Ω·cm) can be lowered by the filling material (approximately 10^−2^ to 10^−3^ Ω·cm). In addition, the electrical conductivity slightly increased when CFs recycled from SHS were applied to CFRPs. These results indicate that SHS recycling enhances interfacial adhesion. Further, because the interfacial bonding strength of SHS-RCF was higher than that of virgin CFs, the electrical resistance of CFRPs applied with SHS-RCF decreased; that is, the interfacial bonding force at this stage can reduce the number of point defects and improve the electrical conductivity of CFRPs. Figure 4 is a schematic diagram showing the changes in interfacial adhesion that can occur as recycling methods of the CFs. As a result, it was confirmed that the interfacial adhesion strength was statistically increased due to the cleanliness (degree of recovery) and oxidation of the carbon fiber surface, and the most important factor was the recycling technology that can recover cleanly.

## 3. Materials and Methods

### 3.1. Materials

In this study, the CFs with 4.9 GPa and 230 GPa of tensile strength and modulus, respectively, were supplied by Toray (12 K, T700). The average diameter and density of the CFs were 7 µm and 1.8 g/cm^3^, respectively. The epoxy resin, diglycidyl ether of bisphenol-A (DGEBA, YD-128, Kukdo Chem., Seoul, Korea), with an equivalent weight of 185–190 g/eq and a viscosity value of 11500–13500 cps at 25 °C, was used as a matrix. Diaminodiphenylmethane (DDM, Tokyo Chem., Tokyo, Japan) was selected as a hardener, and methylethylketone (MEK, Daejung Chem., Shiheung, Korea) was used to reduce the high viscosity of DGEBA.

LDPE (XJ800), with 0.914 cm^3^/g density and 103 °C melting temperature, was purchased from Lotte Chemical Corporation (Seoul, Korea).

### 3.2. Sample Preparation

CFs were impregnated with thermosetting resin by the drum winding method and converted into prepregs after removing the solvent for 48 h. CFRPs prepared using the hot press method at 175 °C had 36% resin content.

Supercritical recycling involved heating a mixture of CFRPs and ethanol (1:5 weight ratio) in a sealed reactor vessel above 250 °C for 30 min. Supercritical recycling samples were fabricated at a pressure of 80 bar. The resultant samples were then rapidly washed, and the remaining product was washed with deionized water at least thrice. Subsequently, the recycled CFs were vacuum-filtered and dried in a heating oven at 100 °C for 3 h.

A fixed SiC furnace was used to pyrolyze the CFRP composites. An alumina tube with 1000 mm length and 80 mm inner diameter was horizontally mounted in an electrical resistance furnace (15 kW) and heated to a final temperature of 550 °C. A detachable porcelain crucible containing the sample was placed at the center of the quartz tube. The CFRPs were heated to 550 °C at 10 °C/min in the SiC furnace under H_2_O (liquid flow 2 mL/min) and maintained at the target temperature for 30 min to obtain carbonized CFRPs. Subsequently, the gas flow was switched to air (O_2_-21% under N_2_) at a rate of 200 mL/min, while the temperature was increased to and maintained at 550 °C for 60 min. Subsequently, recycled carbon fibers (RCFs) were obtained after cooling to room temperature.

To produce a composite with a thermoplastic resin, the fillers were cut into a unit size of 1 inch. Composites were prepared by mixing LDPE and CF at a tailored mixing ratio; subsequently, the mixture was melt-blended in a mixing chamber at 130 °C with a screw speed of 70 rpm for 30 min in an internal mixer. The total weight ratio of the matrix to the filler was fixed at 5:1. After melt-blending, each sample was molded by hot pressing using a vacuum bag molding method. The processing temperature, time, and pressure were maintained at 130 °C, 15 min, and 10 MPa, respectively. Table 3 enlists the formulation of various mixing fillers of the CFRPs in the composites.

### 3.3. Characterization of Samples

The morphologies of PE/RCF and PE/CF were investigated using a scanning electron microscope (SEM, AIS2000C, Seron Tech. Inc., Uiwang-si, Korea). To reduce charging during SEM imaging, samples were initially placed on a sample holder and coated with platinum. The base pressure of the analyzer chamber was approximately 5 × 10^−5^ Pa and the acceleration voltage was set to 15 kV.

The thermal conductivity of the composite sample was measured by the transient plane source (TPS) method using a hot disk instrument (TPS2500S, Hot Disk Inc., Gothenburg, Sweden), and the conductivity was measured in the horizontal direction.

A nickel coil wrapped in a polyimide film (Kapton) with a diameter of 6.4 mm (#5501) was used as a probe for measuring the thermal conductivity and thermal diffusivity of the composite sample and to reduce the error caused by the anisotropic fillers. The estimated data reproducibility and accuracy of the equipment provided by the manufacturer were more than 1% and 5%, respectively. In all experiments, the measuring instrument was preheated for 30 min or more to ensure analysis reliability, and the measurement environment was maintained at 25 °C and 30% relative humidity.

All specimens were pressed flat using an automatic polishing device at intervals of 5 min using sandpapers of 1000, 2000, and 4000 grades. The samples were placed approximately 6 cm from the center of the automatic polishing machine, and a force of 50 N and a speed of 150 rpm were applied uniformly across the samples. Finally, the finished samples were uniformly cut into dimensions of 30 × 30 × 3 mm.

The electrical resistivity of the samples was measured using a Loresta GP resistivity meter (MCP-T610, Mitusbishi Chemical Co., Tokyo, Japan) connected to a 4-point-probe (MCP-TP03P, Mitsubishi Chemical Co., Tokyo, Japan), which was used to eliminate the effect of contact resistance. Further, at least 10 samples were tested for the reliability of each formulation.

### 3.4. Measurement of Physical Properties

An impact strength test was conducted to observe the changes in the mechanical strength of the composites having the fillers. A Charpy (CEAST^®^ Resil Impactor, CEAST, MA, FL, USA) pendulum impact test was employed according to ASTM D6110 [30] to examine the total required energy until the final fracture of the composite material.

## 4. Conclusions

This study compared the characteristics of CFs/polymer composites acquired through different recycling methods. The results indicated that pyrolysis and supercritical recycling methods were efficient. The recycled fiber “PE/SHS-RCF” exhibited an impact strength of 106.88% compared with the commonly used fiber (T700). Further, the thermal conductivity of the CFs was as high as 106%. Moreover, the formation of oxygen functional groups on the CF surface during pyrolysis increased the IFSS between the resins. Lastly, the supercritical H_2_O/CO_2_ process showed the highest energy efficiency and, thus, the further modernization and development of this process on a large production scale can facilitate its application in domestic recycled CF markets.

## Figures and Tables

**Figure 1 molecules-27-05663-f001:**
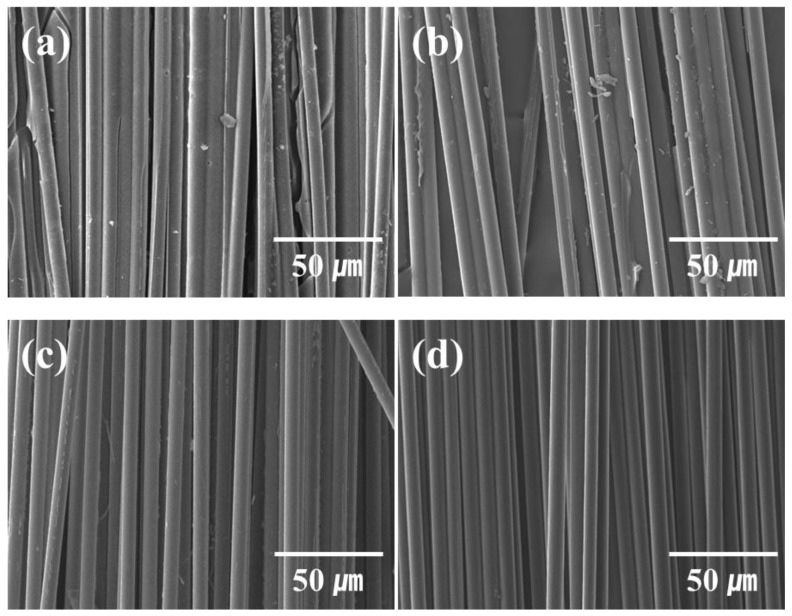
SEM images of waste and recycled carbon fiber (CF) acquired through different recycling methods. (**a**) CFRP scrap wastes, (**b**) recycled CFs from supercritical method, (**c**) recycled CFs from superheated steam pyrolysis, and (**d**) virgin CFs.

**Figure 2 molecules-27-05663-f002:**
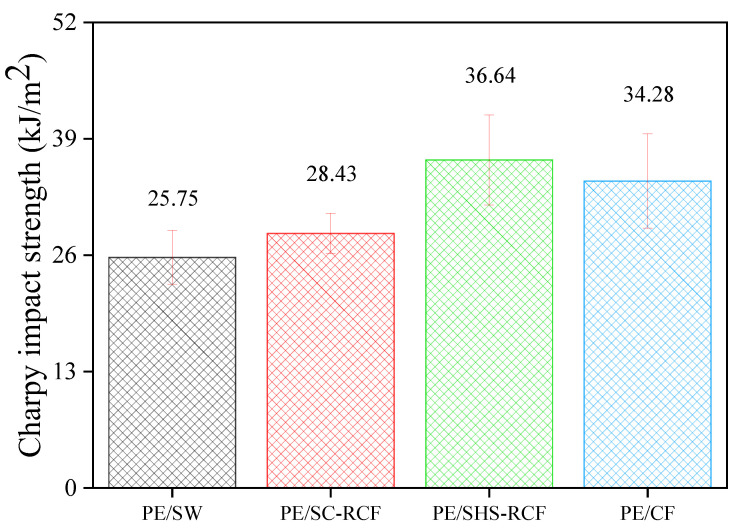
Charpy test results of the impact strength of RCF-reinforced low-density polyethylene (LDPE).

**Figure 3 molecules-27-05663-f003:**
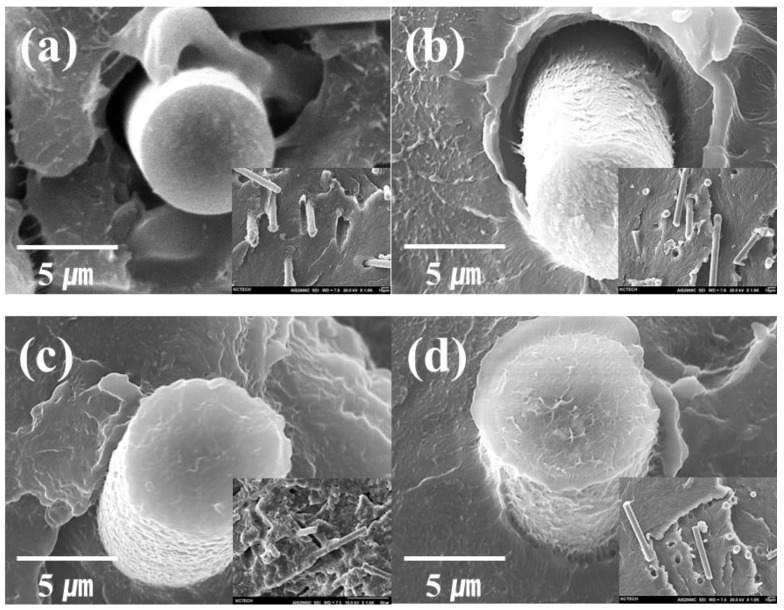
SEM images of the fractured surfaces of polymer composites with CF types acquired from the Charpy pendulum impact test. (**a**) PE/SW, (**b**) PE/SC-RCF, (**c**) PE/SHS-RCF, and (**d**) PE/CF.

**Figure 4 molecules-27-05663-f004:**
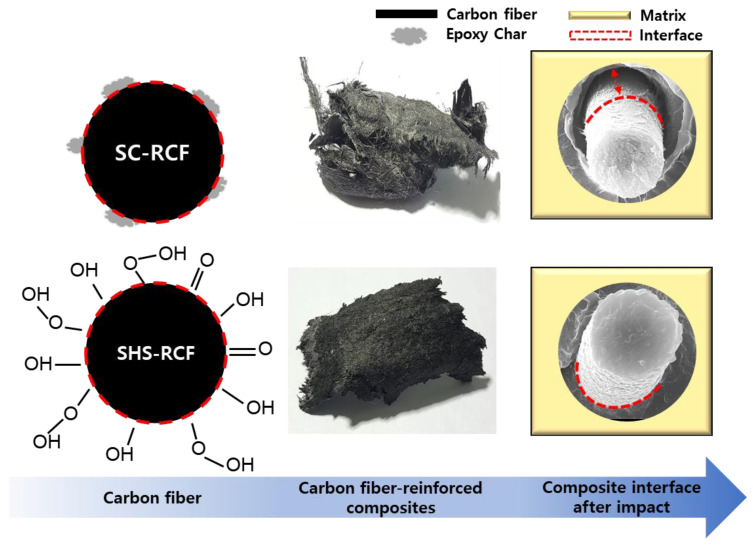
A schematic of interfacial adhesion changes of carbon fibers by recycling methods.

**Table 1 molecules-27-05663-t001:** Summary of the results of previous studies according to carbon fiber type (B. J. Kim et al.) [23].

Property	Units	Sample Name		
		SC-RCF	* SHS-RCF	* CF(T700)
Tensile strength	GPa	3.42	3.88	4.28
Interfacial shear strength	MPa	33.62	47.06	39.19
Oxygen content	%	13.37	14.29	8.79
Polarity value	mN/m	-	12.52	5.66
Surface free energy	mN/m	-	35.35	27.77

* B. J. Kim et al., Journal of Environmental Management, 203, 872–879 2017 [23].

**Table 2 molecules-27-05663-t002:** Thermal and electrical conductivity of polymer composites with different carbon fiber types.

Property	Units	Sample Name	
		PE/SHS-RCF	PE/CF
Heat conductivity	W/mK	1.87	1.76
Resistivity	Ω·cm	3.2 × 10^−3^	1.5 × 10^−2^

**Table 3 molecules-27-05663-t003:** CFRP formulation for composite preparation.

Sample Name	Filler Type	LDPE (g)	Filler (g)	Temperature (°C)	Mix (rpm)	Time (min)
PE/SW	CFRP scrap wastes	150	30	130	70	30
PE/SC-RCF	Recycled CFs(supercritical)					
PE/SHS-RCF	Recycled CFs (superheated steam)					
PE/CF	T-700 CFs					

## Data Availability

Not applicable.

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
