# Peer review of "Comparison of the Characteristics of Recycled Carbon Fibers/Polymer Composites by Different Recycling Techniques"

_molecules, 2022, doi:10.3390/molecules27175663_

Round 1

Reviewer 1 Report

In this manuscript, the authors compared different recycling methods on the waste carbon fiber-reinforced plastic, including mechanical recycling, chemical processes, and thermal processing. The authors compared morphologies, thermal properties, electrical properties, and mechanical properties of different carbon fibers. This paper is interesting. However, some critical measurements are missing, and some analysis cannot be supported without providing experimental results. I recommend a major revision on this manuscript. Below are comments.

(1)    There are many grammar issues throughout the manuscript. A careful proofread is required.

(2)    The authors didn’t measure mechanical properties of recycled carbon fiber in their own experiment. Although they claimed the experiment conditions are the same, it is suggested to do their own measurement.

(3)    How did the authors collect the average IFSS value for CFs?

(4)    The authors discussed extensively on surface morphologies, but only provided a small image at the corner. The scale bar in each image is hard to read.

(5)    The authors should provide XPS result to support their explanation on the enhanced IFSS values.

(6)    The authors only compared the thermal and electrical properties between PE/SHS-RCF and PE/CF. Why PE/SC-RCF sample is missing in this measurement?

Author Response

Thank you for your kind review by reviewers.
I have carefully written it and attached the file. Please review.

Reviewer 2 Report

This manuscript compared three kinds of recycled carbon fibers (RCFs) from mechanical grinding, steam pyrolysis, and supercritical solvent process for their application in synthesizing polymer-matrix composites, and it was well-organized. The article can be considered to be published on Molecules. Meanwhile, I suggest the authors to add the surface chemistry of RCFs by XPS or FTIR in order to strongly support the explanations and conclusions.

Author Response

(The authors gave the same response as above.)

Reviewer 3 Report

I would like to thank the authors for their nice contribution and well-written paper. The present referee is an expert in carbon fiber manufacturing and characterization.
The present paper gives a clear overview of the impact of different recycling methods of carbon fibers on the mechanical performance of the resulting CF thermoplastic composites. 
The following points are needed to give better insights into the structure and molecular evaluation on why the SHS process leads to better mechanical performance:

1- i suggest the authors to use XPS to evaluate the molecular origin of the better wettability of the fiber surfaces after recycling using different techniques, especially in the case of the SHS method
2- i suggest the authors to measure the surface tension of the virgin fibers and the recycled fibers
3-  i suggest the authors to give more details regarding the Charpy test they performed. 

I recommend the journal accept the paper once the authors provide more details regarding my comment under the above-mentioned point 2, as it is very essential to evaluate the functionality of the fiber surface. 

With the best wishes!

Author Response

(The authors gave the same response as above.)

Round 2

Reviewer 1 Report

The revision looks good to me. I suggest to accept this manuscript.

Reviewer 2 Report

I am satisfied with the manuscript in its current form and I recommend it for publication.